# Retrieval of Photometric Parameters of Minerals Using a Self-Made Multi-Angle Spectrometer Based on the Hapke Radiative Transfer Model

Ping Zhou [1], Zhe Zhao [1,2], Hong-Yuan Huo [3,*] and Zhansheng Liu [3]

1. School of Geosciences and Resources, China University of Geosciences (Beijing), Beijing 100083, China; zhoupx@cugb.edu.cn (P.Z.); zhaozhe@hebmt.gov.cn (Z.Z.)
2. Hebei Bureau of Coal Geological Exploration, Shijiazhuang 050085, China
3. Faculty of Architecture, Civil and Transportation Engineering, Beijing University of Technology, Beijing 100124, China; liuzhansheng@bjut.edu.cn
* Correspondence: huohongyuan@bjut.edu.cn; Tel.: +86-18610564621

**Abstract:** In this paper, a self-made, mineral, multi-angle, spectrum measurement device is employed to measure the multi-angle spectra of olivine and plagioclase; the multi-angle spectra of ilmenite in the Reflectance Experiment Laboratory (RELAB) Spectral Library are collected; and the optimized retrieval of the photometric parameters of the Hapke model is realized. Importantly, the derived result of the single-scattering albedo (SSA) is stable and has both mathematical meaning and physical meaning. The derived Legendre polynomial coefficients of the phase function can better simulate the variation in the mineral spectra with angle. This paper compares the effects of multi-angle and single-angle spectral data on the photometric parameter derived results. The setting of the Legendre polynomial coefficient of the scattering phase function mainly affects the simulation accuracy of the mineral spectra as a function of angle. Using this coefficient to optimize the retrieval, the simulation accuracy is moderately improved compared with the single-angle simulation. The estimation of photometric parameters based on multi-angle spectral data can provide a basis for setting the empirical values of the phase function parameters from single-angle spectral calculations, which can more truly reflect the law of reflectance spectra changing with angle than Lucey's traditional empirical value of the phase function (b = −0.4 and c = 0.25). The results of multi-angle spectra retrieval in this paper show that the Legendre polynomial coefficients of the phase function vary with wavelength rather than being constant and that different minerals differ greatly.

**Keywords:** Hapke model; optimized retrieval; photometric parameters; optical constants

## 1. Introduction

When light enters a mineral, part of the light is absorbed, and part of the light is reflected, showing spectral features that are dependent on internal and external factors unique to a sample. This behavior is the result of the combined effect of the internal and external conditions of the mineral [1,2]. Studying the variation in spectral features over a range of physical, compositional, and experimental parameters is important for interpreting the internal attributes and external conditions implicit in the spectra [3–6]. The spectral characteristics of minerals are mainly determined by the crystal field effect, charge migration, conduction band, color center, group vibration, and other factors within the constituent materials [7,8]. They are also affected by the size of the mineral particles, mineral mixing ratio, degree of weathering, and observation geometric conditions. Studying the factors that influence spectral character and deconvolving the influences of the various internal aspects and external factors are important research areas that facilitate the extraction of mineral information and derivation of mineral abundances from spectra of natural samples [1,5,6].

Optical constants are the basic material properties that describe the propagation of light within a particulate medium and are independent of the size and shape of minerals [9,10]. They are the real and imaginary parts of the complex index of refraction ñ = n+ik, where the real part describes the refraction of light and the imaginary part describes the absorption of light. The refractive coefficient and absorption coefficient characterize the refraction ability and absorption ability, respectively, of electromagnetic radiation when light enters the interior of minerals. However, the optical constants of most of the major rock-forming minerals are either difficult to measure or not readily available [11–14], as these constants are difficult to measure directly in the laboratory; on the other hand, these measurements require a large number of high-quality samples, especially for minerals with weak absorption features [9].

The methods for determining the optical constants can be divided into three groups: transmission spectroscopy models [15], smooth surface reflectometry models [16], and powder surface reflectometry models [17]. The transmittance spectroscopy method for deriving optical constants requires a measurement of the thickness of the transmissive mineral slices, but this measurement is difficult to carry out [15]. The smooth surface reflectometry method is used to derive the optical constants of minerals based on the Fresnel reflection law by measuring the specular and polarized reflectance spectra of the smooth surfaces of minerals [17–19]. This method is widely employed for mid-infrared and thermal infrared spectra. The powder surface reflectometry method is used to calculate the single-scattering albedo (SSA) and other photometric parameters of the minerals based on the Hapke radiative transfer model through measurement of the bidirectional reflectance spectra of the powder surface of minerals. Under the condition of setting the refractive index, the absorption coefficient of the mineral is calculated based on the derived SSA and its relationship with the refractive index and absorption coefficient [8].

The radiative transfer model is an important method for interpreting information with respect to the mineral composition included in visible and near-infrared (VIS-NIR) light [20]. The Hapke semi-empirical bidirectional reflectance model quantitatively describes the physical properties of the electromagnetic radiative interacting with a semi-infinite particle medium [21–27]. Hapke pointed out that reflectance is a function of the SSA. As an important photometric parameters, the SSA is irrelevant to the observation angle and can be linearly mixed according to the volume percentage of mineral end-members. Furthermore, the SSA is a function of the optical constants. The Hapke radiative transfer model has been shown to be effectively applied to spectral decomposition and content retrieval of mixed minerals [28,29] and to estimate the roughness, particle size, porosity and other physical properties of weathered surfaces [28]. Lucey (1998) applied the Hapke model to successfully derive the optical constants of olivine, low-calcium orthopyroxene and high-calcium clinopyroxene and pointed out that the reflectance spectra of 0.4–2.5 μm were a function of mineral composition and particle size [9]. Denevi et al. (2008) investigated the mixing of orthopyroxene and olivine and the mixing of orthopyroxene and clinopyroxene [20]. The Hapke model was shown to be feasible for both simulating the mixed spectra of known mineral components and deriving the mineral components by using the mixed spectra of known minerals.

Yang et al. (2018) showed that the phase function parameter in the photometric parameter is wavelength-dependent [30]. The Legendre polynomial coefficient of the scattering phase function mainly refers to the approximate average value of the mineral forward scattering coefficient of Mustard and Pieters (1989); b is generally set to −0.4; and c is generally set to 0.25 [9,31]. However, these data only cover the wavelength range of 0.6–1.6 μm. Pilorget et al. (2016) discovered many diagnostic absorption features of planetary surface minerals in the wavelength range of 1.6–2.5 μm, such as the absorption of $Fe^{2+}$ in pyroxene and spinel [32]. The charge transfer between the elements $Fe^{2+}$ and $Ti^{4+}$ will cause a stronger absorption of the spectrum in the ultraviolet (300–600 nm) band, and the shorter the wavelength is, the stronger the absorption, causing the spectrum to become more "red" (from ultraviolet light to visible light, as the wavelength increases, the reflectivity increases).

The $Fe^{2+}$ in the spectrum will also cause absorption characteristics at approximately 1000 nm. The increase in ilmenite will weaken the "red" degree of the spectrum, making the absorption characteristics increasingly weaker. Ilmenite is a highly absorptive opaque mineral that exhibits a significantly different scattering behavior from transparent or translucent silicate minerals such as olivine, pyroxene, and plagioclase. Ilmenite mainly exhibits backscattering behavior, while silicate minerals show more forward scattering behavior. Robertson et al. (2017) showed that ilmenite-bearing mixtures are spectrally unmixed in the VIS-NIR range; if ilmenite uses a phase function similar to silicate minerals, Legendre polynomial parameters will produce large uncertainties in the abundance of simulated minerals [33].

Recently, the single-angle reflectance spectroscopy dataset has mostly been employed in research on the derivation of photometric parameters and optical constants of minerals based on the Hapke radiative transfer model. However, when performing photometric parameter and optical constant derivation based on single-angle spectral data, assumptions about the empirical values of photometric parameters (such as the phase function) other than SSA must be made, which will affect the accuracy of the results to a certain extent. The use of multi-angle reflection spectrum data can optimize the derivation of photometric parameters, including single scattering albedo and phase function parameters. This paper attempts to discuss the advantages of multi-angle reflectance spectra over single-angle reflectance spectra for optimizing the derivation of photometric parameters and optical constants. A comparison analysis of their difference in the simulation of spectra of minerals based on the derived optical constants and photometric parameters is performed to further show their performance on the retrieved results.

## 2. Self-Made Instrument of Multi-Angle Spectrometer Measurement

The self-made, multi-angle spectrometer consists of two turntables and two brackets (refer to Figure 1). The larger turntable is used as the base to support the smaller turntable and two brackets. The light source and optical fiber probe (Analytical Spectral Devices (ASD) FieldSpec Pro FR, manufactured by ASD, Inc., Boulder, CO, USA, with a wavelength ranging from 350 nm to 2500 nm) are installed on the two brackets. By rotating two brackets and a smaller turntable, multi-angle spectra under different geometric conditions can be theoretically measured. In practical measurements, due to collisions between the probe and the light source, light shielding and decreased spectral stability with a large incident angle, the measurable angle is limited. Due to the collision between the probe and the light source, light blocking, and when the incident angle is large, the stability of the spectrum measurement decreases, and the measurable angle is limited. Currently, The incident angle (the angle between the light source and the normal vector of the surface) and the emittance angle (the angle between the probe and the normal vector of the surface) vary between 40° and 90°, the relative azimuth angle between the probe and the light source varies between 15° and 245°, and the phase angle between the probe and the light source varies between 15° and 80°. Considering the effect of shadows caused by light being blocked, the possible collision between the light source and the optical fiber probe, and the influence of the decrease of the stability of the spectrum measurement produced by the container (wall) holding the sample, the phase angle range we measured is from 20 to 60°.

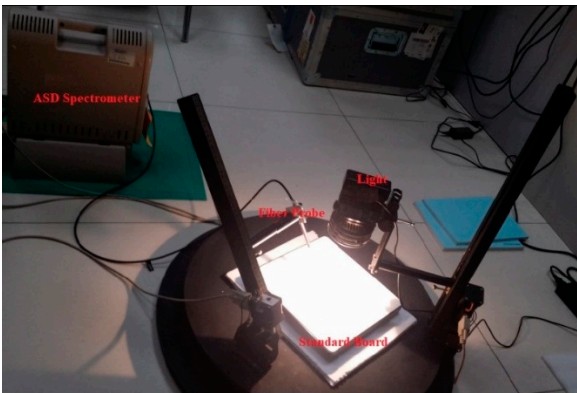

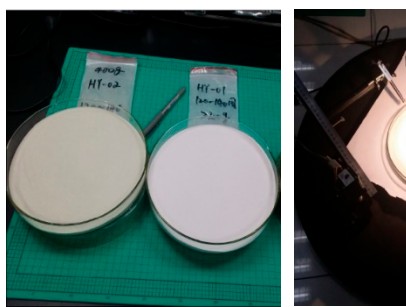
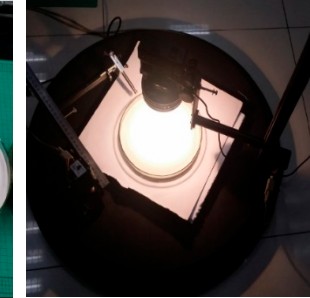

**Figure 1.** Measuring equipment for the multi-angle spectral data.

## 3. Materials Collection and Methodology

### 3.1. Multi-Angle Spectra Data Collection

Currently, the photometric parameters of these minerals with different reflectances in most of the literature generally adopt unified empirical values, which reduces the accuracy of spectral simulation, although some researchers have made assumptions about the photometric parameters in the absence of data on the minerals of interest. This study is based on a self-made, mineral, indoor, multi-angle spectrum measurement device (Figure 1). Three sample minerals—plagioclase, olivine, and ilmenite—are selected, and their multi-angle spectrum curves are collected. These three typical rock minerals are selected mainly considering that the reflectivity of the three minerals of plagioclase, olivine and ilmenite are ranked from high to low, representing light-colored minerals (high-reflectivity minerals), medium reflectivity minerals, and dark minerals (low reflectivity minerals). The purpose of selecting the three minerals with very large differences in reflectivity is to analyze the difference in the photometric parameters of minerals with different reflectivities and to provide a more accurate basis for the setting of photometric parameters during spectrum simulation.

This spectral collection experiment was carried out on 22 January 2015 at the China Aero Geophysical Survey and Remote Sensing Center for Land and Resources, Beijing, China. The reflectance spectra of olivine and plagioclase at 15 test angles (Table 1) were measured by using an ASD spectrometer and a multi-angle spectrometer, respectively. During the experiment, the built-in light source of the ASD spectrometer served as the light source, and the ASD built-in optical fiber probe served as the spectrum probe. According to the calibration accuracy of the spectrometer, the main parameters, such as spectral resolution, center wavelength position, and signal-to-noise ratio, were calibrated and tested. After the instrument starts and reaches the warm-up time of the instrument, it initializes the dark current measurement. During the measurement, the dark current of the spectrometer is measured every half an hour to correct the influence of instrument noise on the measurement results in time. A standard white board was placed horizontally and the probe was aimed at it to optimize the instrument parameters. The probe was aimed at the target to be measured, and the target reflection spectrum was collected.

**Table 1.** Configuration table of the measured angle for the multi-angle spectral data.

| ID | Incident Angle i | Azimuth φ | Emergence Angle e | ID | Incident Angle i | Azimuth 0φ | Emergence Angle e | ID | Incident Angle i | Azimuth φ | Emergence Angle e |
|----|-----|-----|----|----|-----|-----|----|----|-----|-----|----|
| 1 | | | 20 | 6 | | | 20 | 11 | | | 20 |
| 2 | | | 30 | 7 | | | 30 | 12 | | | 30 |
| 3 | 0 | 180 | 40 | 8 | 0 | 210 | 40 | 13 | 0 | 240 | 40 |
| 4 | | | 50 | 9 | | | 50 | 14 | | | 50 |
| 5 | | | 60 | 10 | | | 60 | 15 | | | 60 |

During the experiment, here, the correction of the spectra for irregularities in the 2–24 micron region of Spectralon was not carried out for this experiment. Due to the collision between the probe and the light source, light blocking, and when the incident angle is large, the stability of the spectrum measurement decreases, and the measurable angle is limited. The samples of minerals, including plagioclase and olivine, utilized in this study were purchased by China Aero Geophysical Survey and Remote Sensing Center for the Land and Resources of China Geological Survey. All the samples are referenced to SpectraLon before they are measured. The sample cup is 2 cm deep and 20 cm in diameter. The fiber probe is 1 cm away from the sample surface during measurement. We could not through the sample to the bottom of the cup, and we did not have spectral contributions from the side of the sample cup—especially at high angles. The mineral powder samples and sample cups were sealed and stored in the laboratory. During the measurement, the mineral powder samples were evenly spread over the sample cup; the surface was flat, and the effect of "hot spots" was almost negligible. The spectra of ilmenite were collected from the Reflectance Experiment Laboratory (RELAB) spectra laboratory. The grain size of minerals of plagioclase and olivine was measured by a Mastersizer 2000 (with Malvern Instruments Co. Ltd., Malvern, UK) at the State Key Laboratory of New Ceramics and Fine Processing, School of Materials Science and Engineering, Tsinghua University on 18 May 2015 and 19 May 2015. The average values of the measured grain sizes of the minerals plagioclase and olivine were 125.8 µm and 131.1 µm, respectively (See Figures 2 and 3).

Before measuring the mineral spectrum with the ASD spectrometer, first, we aligned the light source and the optical fiber probe to the Spectralon for optimization. The Spectralon is made of barium sulfate ($BaSO_4$) and other materials by pressing or sintering. It is close to Lambertian when the zenith angle is ≤45°. In the actual multi-angle spectrum collection of olivine and plagioclase, the specific settings of the incident angle i, azimuth angle φ, and emergence angle e are shown in Table 1. The incident angle i is set to 0°; the azimuth angle φ is set to 180°, 210°, and 240°; and the multi-spectral curves of olivine and plagioclase at emergence angles e of 20°, 30°, 40°, 50°, and 60° are collected. Here, the phase value is calculated based on the assumption that g = arccos (cos $(\theta i)$*cos$(\theta e)$ + sin$(\theta i)$*sin$(\theta e)$*cos$(\varphi i)$). The choice of incidence angle is unfortunate as sin (0) = 0, which means that the last term will always be zero and that these measurements are independent of the azimuthal angle. Thus, each data set is actually two replicates of the same data set. To solve this problem, the three group spectral datasets with different azimuth angles were processed, the mean spectral curves were obtained by averaging the three group datasets, and then the mean value was applied to retrieve the optical parameters. The reflectance spectral curves are simulated based on these retrieved optical parameters and the phase angle information, including the incident angle, azimuth angle, and emergence angle. A comparative analysis was performed between the simulated spectra and the average spectral curves.

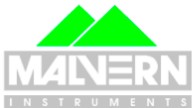 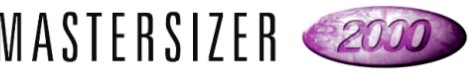

## Grain Size Analysis Report

Sample Name:
OLV120-140

Sample Source/Type:

Reference Sample Batch:

SOP Name:

Operator:
aaa

Source of Results:
Measurment

Measure Time:
2015　5　18　10:31:59

Analysis Time:
2015　5　18　10:32:00

Particle Name:
Mixture
Refractive Index of Particle:
1.500
Dispersant:
Water

Injector Name:
Hydro 2000MU (A)
Particle Absorption Rate:
0
Refractive Index of Dispersant:
1.330

Analysis Mode:
Universal
Particle Size Range:
0.020　to　2000.000　um
Residual:
1.548　%

Sensitivity:
Normal
Shading:
4.55　%
Result Simulation:
Off

Concentration:
0.0862　%Vol

Specific Surface Area:
0.0458　m^2/g

Span:
0.930

Surface Area Average Particle Size D[3,2]:
131.063　um

Consistency:
0.288

Volume Average Particle Size D[4,3]:
147.311　um

Result Type:
Volume

d(0.1):　88.848　um　　　　　　d(0.5):　138.827　um　　　　　　d(0.9):　217.896　um

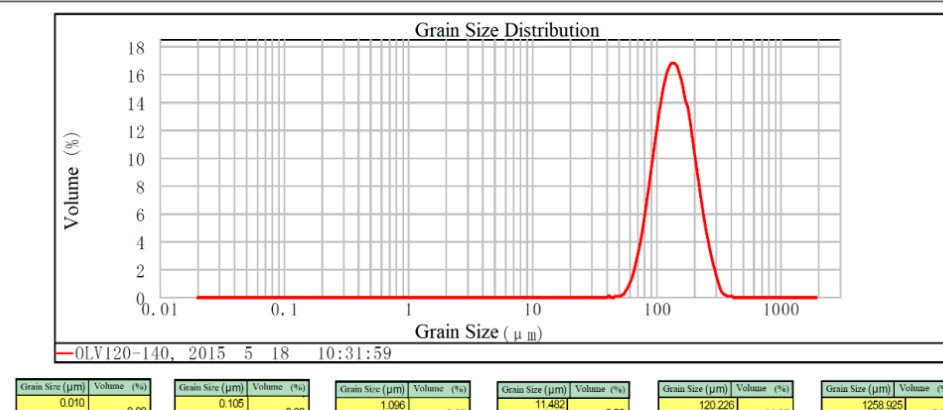

**Figure 2.** The grain size analysis report of olivine.

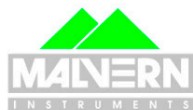
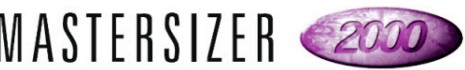

Grain Size Analysis Report

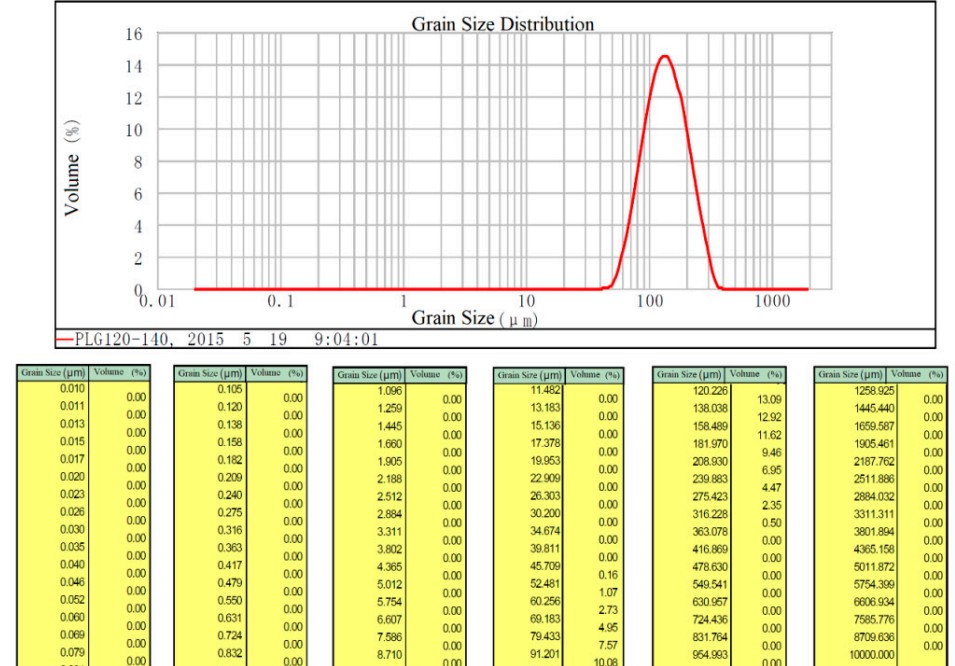

**Figure 3.** The grain size analysis report of plagioclase.

The collected olivine spectral curves are shown in Figure 4, where there are five sets of spectral curves with an incident angle of 0°, an azimuth angle of 180°, and an emergence angle of 20°/30°/40°/50°/60° (Figure 4a). There are five sets of spectral curves with an incident angle of 0°, an azimuth angle of 210°, and an emergence angle of 20°/30°/40°/50°/60° (Figure 4b). There are five sets of spectral curves with an incident angle of 0°, an azimuth angle of 240°, and an emergence angle of 20°/30°/40°/50°/60° (Figure 4c). Five sets of measured spectral curves with an incident angle of 0° and an azimuth angle of 180° and five sets of spectral curves with an incident angle of 0° and an azimuth angle of 240° were employed for the retrieval of olivine photometric parameters. Based on the retrieved photometric parameters of olivine, the multi-angle spectral curves of olivine with an incident angle of 0° and an azimuth angle of 210° are simulated. These

simulated multispectral curves are compared and analyzed with the five measured sets of spectral curves with an incident angle of 0° and an azimuth angle of 210°.

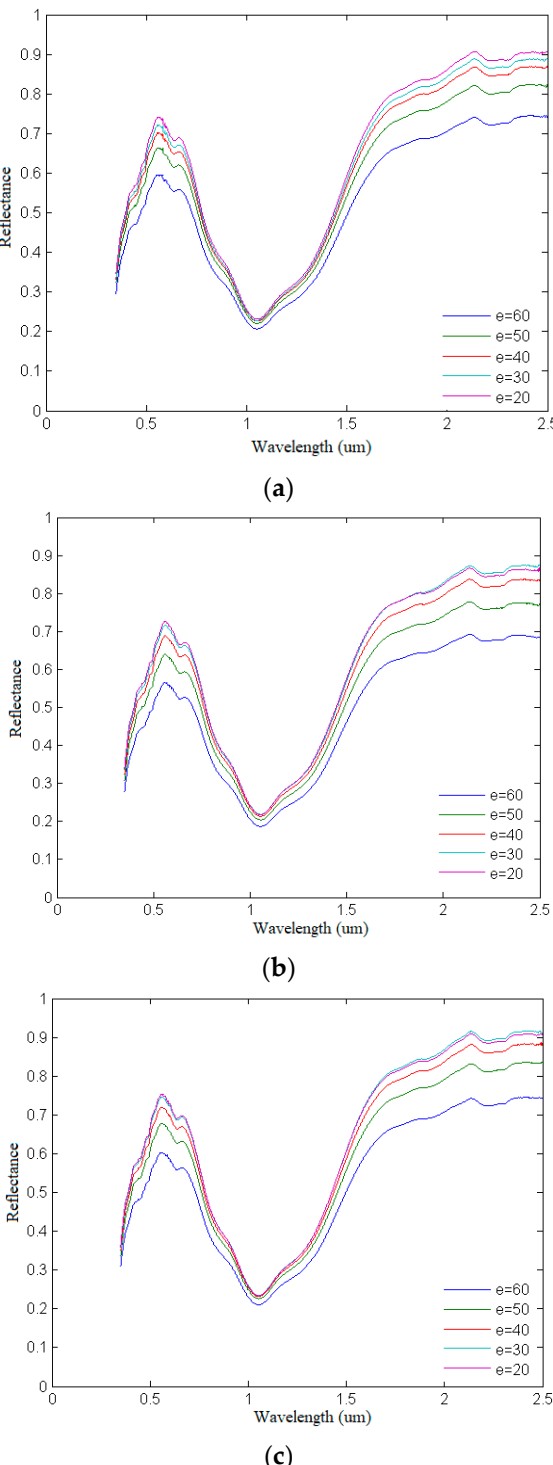

**Figure 4.** Measured multi-angle spectral data of olivine (i.e., incidence angle, emittance angle and azimuth angle), including five multi-angle spectra at an incident angle of 0° and azimuth angle of 180° (**a**); five multi-angle spectra at an incident angle of 0° and azimuth angle of 210° (**b**); and five multi-angle spectra at an incident angle of 0° and azimuth angle of 240° (**c**).

Similar to the olivine spectrum collection, for the spectrum curve of plagioclase with an incident angle of 0°, azimuth angles of 180°, 210° and 240°, and different exit angles of

20°/30°/40°/50°/60°, a total of 15 groups were collected (as shown in Figure 5a–c). Five sets of measured multispectral curves with an incident angle of 0° and an azimuth angle of 180° and five sets of multispectral curves with an incident angle of 0° and an azimuth angle of 240° were employed for the retrieval of plagioclase photometric parameters. Based on the retrieved photometric parameters of plagioclase, a comparative analysis was performed between the simulated multi-angle spectrum curve of plagioclase with an incident angle of 0° and an azimuth angle of 210° and five measured sets of spectral curves with an incident angle of 0° and an azimuth angle of 210°.

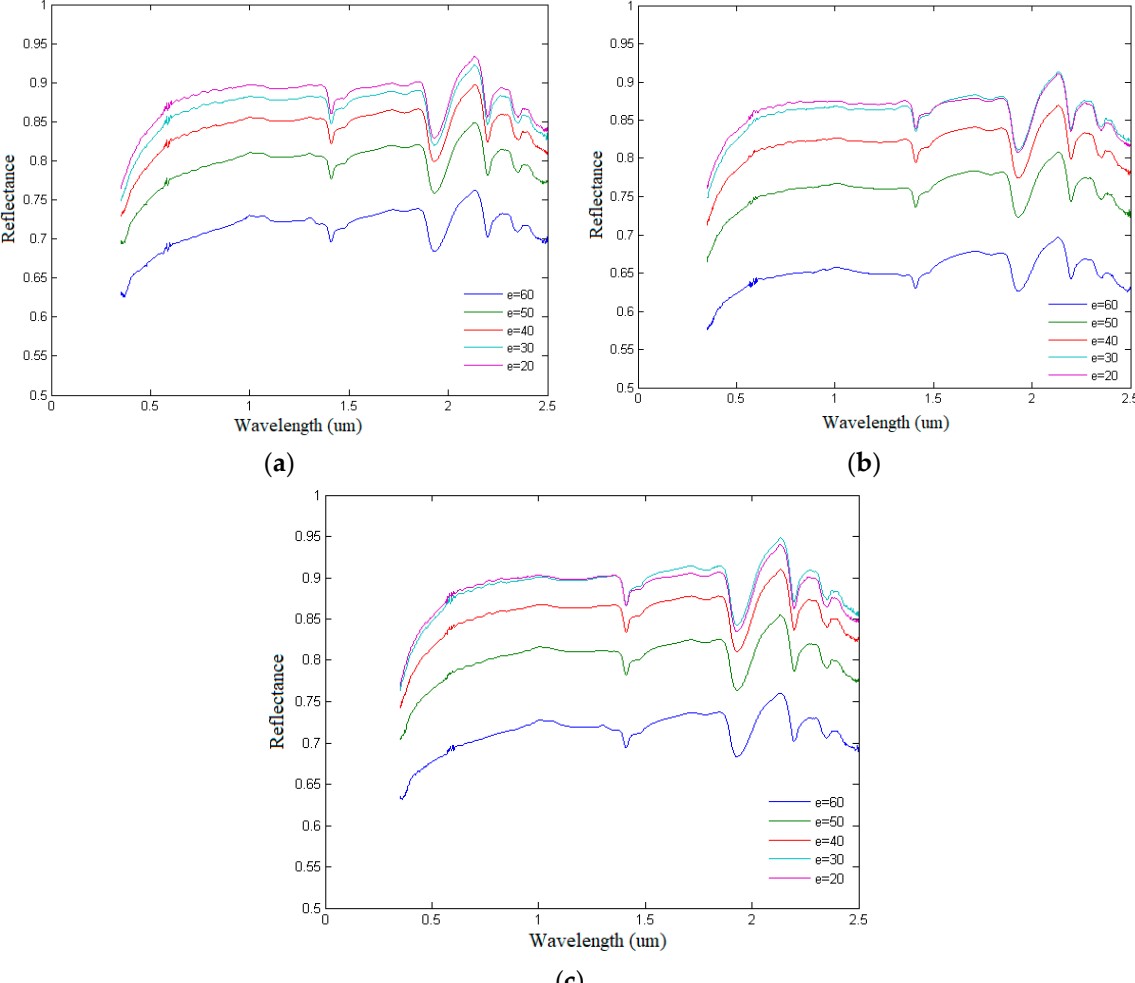

**Figure 5.** Measured multi-angle spectral data of plagioclase (i, e and φ are the incidence angle, emittance angle and azimuth angle, respectively) including five multi-angle spectra at an incident angle of 0° and azimuth angle of 180° (**a**); five multi-angle spectra at an incident angle of 0° and azimuth angle of 210° (**b**); and five multi-angle spectra at an incident angle of 0° and azimuth angle of 240° (**c**).

In addition, the multi-angle spectroscopy data of ilmenite from Brown University's RELAB were collected (Figure 6). The collected ilmenite spectrum data number is MR-MSR-005, the wavelength range is 300–2600 nm, and the spectral resolution is 10 nm. When the exit angle is 0°, the azimuth angle is 0°—unchanged—and the incident angle is 10°, 20°, 30°, 40°, and 50°; a total of five sets of multi-angle reflection spectrum curves are measured.

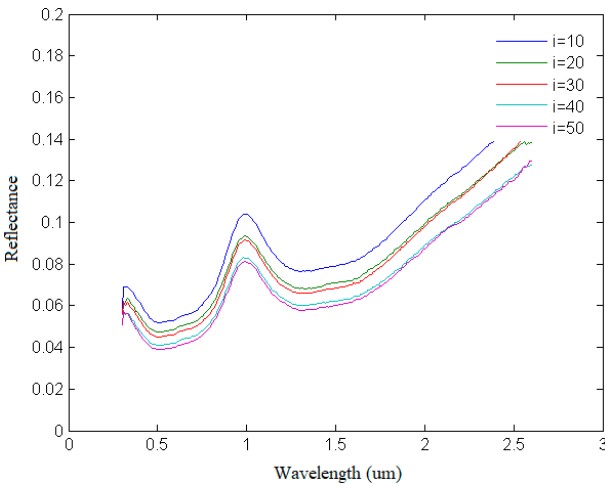

**Figure 6.** Measured multi-angle spectral data of ilmenite (i.e., incidence angle, emittance angle and azimuth angle).

## 3.2. Models and Methodology

Figure 7 shows the flowchart of this study. In this paper, the multi-angle spectrum data of olivine and plagioclase were collected using a self-made, multi-angle, spectrum measurement device, and the multi-angle spectrum data of ilmenite in the RELAB Spectral Laboratory were collected. These collected multi-angle spectrum datasets were processed. Using the Monte Carlo method, the luminosity parameter SSA of the mineral and the Legendre polynomial parameters *b* and *c* of the phase function were derived based on Equations (1)–(13). The simulated spectrum was obtained accordingly. A comparative analysis was then performed between the simulated spectrum and the average measured spectrum.

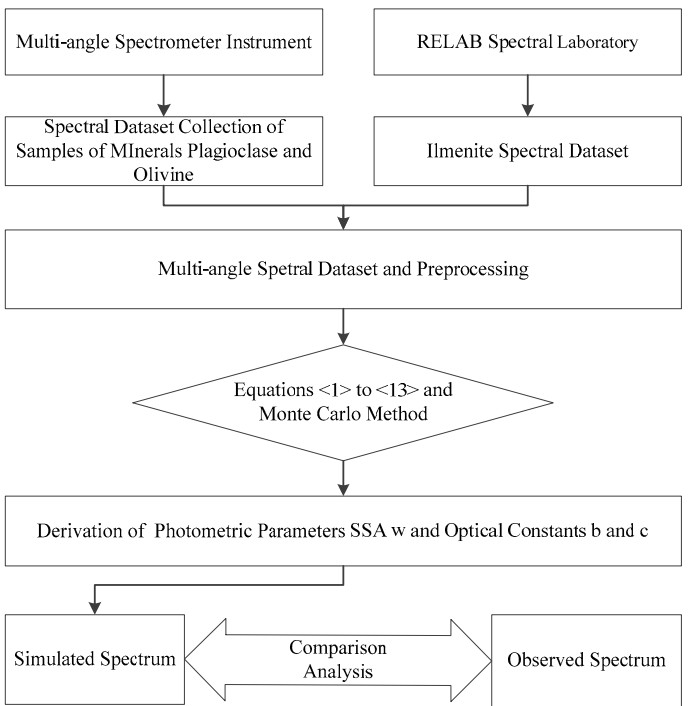

**Figure 7.** Flowchart of the study.

### 3.2.1. Hapke Radiative Transfer Model

Our work is mainly based on Hapke's semi-empirical bidirectional reflectance model [18], which quantitatively describes the physical properties of electromagnetic radiation interacting with a semi-infinite particle medium. Hapke's model is applicable in a particle medium whose particle size is much larger than the wavelength of light. The reflectance of the particle medium can be divided into single scattering and multiple scattering:

$$r(\mu_0, u, g) = \frac{w}{4} \frac{\mu_0}{\mu_0 + \mu}[(1 + B(g))p(g) + M(\mu_0, \mu)] \tag{1}$$

where $\mu_0$ and $\mu$ are the cosine of the incidence angle and emittance angle, respectively, and $g$ is the phase angle. As an SSA, $w$ depends on the wavelength, representing the combined contribution of particles in varieties of sizes per unit volume. $w$ is independent of the viewing geometries and can be mixed linearly by the volume percentage of end-member minerals.

$p(g)$ describes the distribution of the scattering energy at different angles, reflecting the scattering properties of the particle medium. This distribution is a function of the phase angle and wavelength, regardless of the condition of the particle surface. The second-order Legendre polynomial equation of $p(g)$ is given by Mustard and Pieters [22]:

$$P(g) = 1 + b\cos(g) + c[3\cos^2(g) - 1]/2 \tag{2}$$

where $b$ and $c$ are often set as $-0.4$ and $0.25$, respectively, which are the approximate average values of the forward scattering minerals of Mustard and Pieters [22].

$B(g)$ is the backward scattering function used to explain the shadow-hiding opposition effect (SHOE) and is a function of the phase angle ($g$), amplitude ($B_o$) and angular width ($h$) of the SHOE:

$$B(g) = \frac{B_0}{1 + \tan(g/2)/h} \tag{3}$$

$B_o$ represents the probability of light reflected by particles at a small phase angle, which is related to the SSA of the particles. The empirical expression of $B_o$ is:

$$B_o \approx e^{-w^2/2} \tag{4}$$

$h$ describes the range of the phase angle in the SHOE, which is related to the filling factor $\varphi$ of the particles:

$$h = -\frac{3}{8}\ln(1 - \varphi) \tag{5}$$

$M(\mu_0, \mu)$ is a multiple scattering function. Hapke proposed the isotropic multiple scattering approximation function (IMSA) based on the assumption that the particle surface reflects isotropically [12,15], where:

$$M(\mu_0, \mu) = H(\mu)H(\mu_0) - 1 \tag{6}$$

Hapke (2002) proposed that the phase function is a function of the scattering of a single particle size (p(g) is the single-particle angular scattering function); after normalization, the rule is as follows: integrate within 0–4π as a fixed value 4π. The second-order Legendre polynomial equation of $p(g)$ is given by Mustard and Pieters [22]:

Integrate this Formula (1) in the range of 0–4π:

$$\begin{aligned}
\int_0^{4\pi} p(g)d\Omega &= \int_0^{4\pi}[1 + b\cos(g) + c(3\cos^2(g) - 1)/2]dg \\
&= \int_0^{4\pi}[1 + b\cos(g) + \frac{3c}{4}\cos(2g) + \frac{c}{4}]dg \\
&= [g + b\sin(g) + \frac{3c}{8}\sin(2g) + \frac{cg}{4}]\Big|_0^{4\pi} \\
&= 4\pi + c\pi
\end{aligned} \tag{7}$$

The results show that the integral of the phase function in the range of 0–4π is not constant to 4π, and the amount of change cπ has nothing to do with the phase angle g.

We divided the calculated variation cπ by 4π to obtain the phase function correction coefficient c/4 and subtract the correction coefficient c/4 from the original second-order Legendre polynomial function to obtain a new phase function. Its expression is:

$$P(g) \ = \ 1 + b\cos(g) + c[3\cos^2(g) - 1]/2 - \frac{c}{4} \tag{8}$$

This Formula (7) is integrated in the range of 0–4π:

$$\begin{aligned}
\int_0^{4\pi} p(g)d\Omega \ &= \ \int_0^{4\pi} [1 + b\cos(g) + c(3\cos^2(g) - 1)/2 - \tfrac{c}{4}]dg \\
&= \int_0^{4\pi} [1 + b\cos(g) + \tfrac{3c}{4}\cos(2g)]dg \\
&= [g + b\sin(g) + \tfrac{3c}{8}\sin(2g)] \, |_0^{4\pi} \\
&= 4\pi
\end{aligned} \tag{9}$$

The improved phase function satisfies the law that the integral within 0–4π is a fixed value of 4π (See Figure 8).

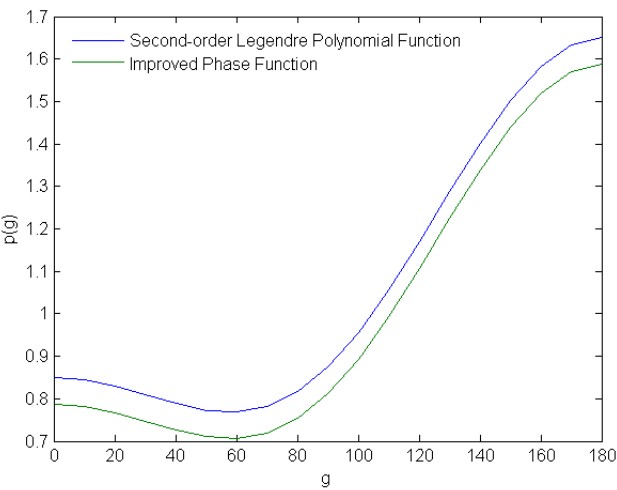

**Figure 8.** The comparison of improved phase function and the second-order Legendre polynomial function.

Hapke discovered that when the SSA is nearly 1.00, the maximum calculation error of the absolute reflectance is 16% by means of the IMSA model, even if the hot effects are not considered. This research also indicated that if the particle media are highly anisotropically scattered, then the absolute accuracy at certain angles is low. Considering these shortcomings, Hapke improved the H-function and determined the multiple scattering function of the anisotropy. The new radiative transfer model based on the anisotropic multiple scattering approximation (AMSA), especially taking into account a new H-function, is able to more accurately express the bidirectional reflectance [16]. The multiple scattering function of anisotropy is given by:

$$M(\mu_0, \mu) \ = \ P(\mu_0)[H(\mu) - 1] + P(\mu)[H(\mu_0) - 1] + p[H(\mu_0) - 1][H(\mu) - 1] \tag{10}$$

$H(u)$ is the H-function in Chandrasekhar's theories [23], and Hapke provides a more accurate analytic approximation of this function [16]:

$$H(x) \ = \ [1 - wx(r_o + \frac{1 - 2r_o x}{2} \ln \frac{1 + x}{x})]^{-1} \tag{11}$$

$$r_o \ = \ (1 - \gamma)/(1 + \gamma) \tag{12}$$

$$\gamma = (1-w)^{1/2} \tag{13}$$

The SSA of the particle medium can be estimated by Equations (1)–(13) if the bidirectional reflectance and viewing geometries are known.

The SSA is a function of optical constants (also referred to as the complex index of refraction). Optical constants, as the basic properties of the material bulk, are the physical quantities that describe the propagation of light interacting with the particle medium, which are independent of the size and shape of minerals [3]. Optical constants are generally expressed as n $= n(1-ik)$, where n and k are the real part and imaginary part, respectively, of the optical constants. n and k characterize the refraction ability and absorption ability, respectively, of electromagnetic radiation when light enters the interior of minerals. In the VIS-NIR spectral range, the n values of the mineral end-members can be considered approximate constants, while the k values of mineral end-members are usually $<< 1$ [3,11,24].

### 3.2.2. Monte Carlo Method

In this paper, the derivation of the photometric parameters adopts a nonlinear optimization algorithm. The largest problem of nonlinear optimization algorithms is that they are highly sensitive to the initial value, that is, different initial input values will produce different optimization results. If the initial value is set unreasonably, it will generate a local optimal solution instead of a global solution. The Monte Carlo method, which is employed to derive the photometric parameters of minerals, is a numerical calculation method based on the theory and method of probability and statistics [34]. This method links the problem to be solved with a certain probability model and uses a computer to realize statistical simulation or sampling to obtain an approximate solution to the problem. Therefore, the Monte Carlo method is also referred to as a random sampling method. Here, the Monte Carlo method can be applied to further overcome the problem of falling into local optimal solutions in model optimization and improve the accuracy of the inversion of the photometric parameters of the Hapke model. The retrieval of the photometric parameters of minerals by the Hapke model has a high accuracy and can be employed as important basic data for spectral simulations. The Monte Carlo method randomly sets a certain number of model initial values within certain constraints and performs optimization solutions, compares all optimization results, and obtains the optimal results.

### 3.2.3. Optimization Retrieval of Photometric Parameters

In this paper, powder surface reflectometry was selected to optimize the derived photometric parameters and single scattering albedo of minerals. The principle of this method is to calculate the SSA and other photometric parameters of minerals based on the Hapke radiative transfer model by measuring the bidirectional reflectance spectra of the mineral powder. In this paper, the Hapke model is used to retrieve the photometric parameters and SSA, and the photometric parameters obtained from the derivation are applied to simulate the spectrum. The simulated spectrum is compared with the measured spectrum to study the error. Here, under the condition of multi-phase angle observation, the same Hapke formula and parameters are used to derive the photometric parameters from the multi-angle spectrum and to simulate the multi-angle spectrum from the photometric parameters.

According to the Hapke radiative transfer model, the SSA is considered to be independent of the phase angle. Thus, the SSA produced by the multi-phase angle spectroscopy is the same, and all phase changes are explained by the phase function coefficients b and c. The photometric parameters that determine the reflectance spectral characteristics of rocks and minerals are the SSA w, SHOE parameter h and second-order Legendre coefficients b and c of the phase function. There is no change in the macro-roughness caused by topographic fluctuations in the laboratory spectral measurement scale of the powder samples, and the correction factor of macro-roughness is not considered in the model. During the process, the LSQCURVEFIT program of MATLAB is called; the multi-phase angle spectrum

data (five sets of data) are utilized as the dependent variable; and the angle information (five sets of data) is employed as the independent variable to derive the unique single SSA and phase function coefficients b and c corresponding to each band.

The filling factor φ is the key parameter to determine the width h of the SHOE. According to previous studies, the SHOE is mainly manifested when the reflection phase angle is less than 15° [22]. The phase angles of the multi-angle spectra adopted in this paper are all greater than 15°, which do not have the ideal angle conditions for the retrieval of the filling factor φ. The Monte Carlo method was used to randomly set the value of the filling factor φ within a certain range, and the photometric parameters of minerals (SSA w, phase function parameters b and c) were retrieved. The parameters with a minimum retrieval error were selected as empirical parameters by Newton interpolation and least-squares estimation. During this Newton interpolation process, the lsqcurvefit function of MATLAB is called to complete Newton interpolation. The calling format is:

$$[a, rnorm, r, exitflag] = lfqcurvefit(fun, a_0, X, Y, l_b, u_b, options)$$

where fun is the expression of the model to be fitted, $a_0$ is the initial estimated value of the model coefficient, and $l_b$ and $u_b$ are the estimated lower bound and estimated upper bound, respectively, of the fitting coefficient. The options for the parameter settings of the fitting process include Maxlter (maximum number of iterations), TolFun (allowable value of the sum of squares of the function parameters), TolX (allowable error value of the fitting coefficient) and Display (control of the display of the fitting process). Among the parameters returned by the function, a is the fitting estimated coefficient, rnorm is the sum of squared errors, r is the residual of the fitted model, and exitflag is the running status. Calculations revealed that the setting of the filling factor φ had a minimal effect on the retrieval error. When the filling factor is 0.35, the retrieval error of the photometric parameters is the smallest, which is the same as the value estimated by Ciarniello [20]. Therefore, the filling factor was set to 0.35, and photometric parameters such as the SSA w and phase function parameters b and c were retrieved.

## 4. Results and Discussion

### 4.1. Retrieval of Photometric Parameters and Spectral Curve Simulation of Olivine

4.1.1. Photometric Parameter Retrieval of Olivine

In this paper, the photometric parameters of olivine, such as SSA w and phase function parameters b and c, were retrieved from the mean spectral dataset, which are obtained by averaging the five multi-angle spectra with an incidence angle of 0° and azimuth of 180° and five multi-angle spectra with an incidence angle of 0° and azimuth of 240°. The curve of the SSA is continuous and smooth with distinct absorption features. The average values of the phase function parameters b and c are −0.9668 and 0.9221, respectively (refer to Figure 9). When SSA, b, and c are derived at the same time, the parameters SSA and b have a better performance than that of parameter c due to its great variation and lesser robustness. The range of b is [−1, 1], which means the front and back scattering ratio coefficient of the phase function. Negative values have more forward scattering and positive values have more back scattering. In Figure 9b, most of the b inversion is negative, which just reflects the forward scattering of silicate minerals. The phase function parameters have physical upper and lower ranges: the c range is [0, 1], which represents the amplitude of the phase function. In Figure 9c, some point values of c almost reach to 1, due to c reflecting the amplitude of the phase function, and here, the upper limit of 1 may be interpreted as stronger scattering. The coefficient b reflects the ratio of front and back scattering, and most b inversions are negative, which reflects the forward scattering of silicate minerals. The coefficient c characterizes the degree of side scattering. The RMSEs of incidence angles of 20°, 30°, 40°, 50°, and 60° are 0.0149, 0.0059, 0.0133, 0.0098, and 0.0144, respectively.

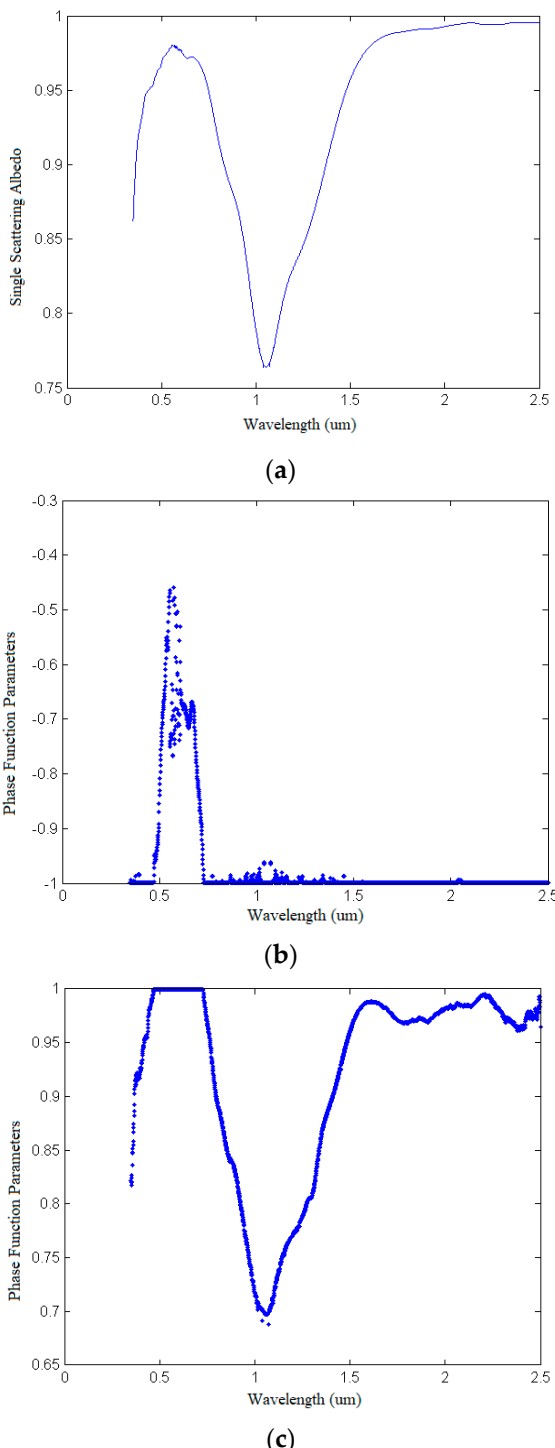

**Figure 9.** Photometric parameters of olivine: (**a**) SSA (w), (**b**) Legendre polynomial coefficients of phase function b, and (**c**) Legendre polynomial coefficients of phase function c.

4.1.2. Spectral Curve Simulation of Olivine Based on Retrieval of Photometric Parameters

Five multi-angle spectra with an incidence angle of 0° and azimuth of 210° were simulated by using the retrieved photometric parameters of olivine (refer to Figure 10). In Figure 9, the multi-angle spectral curves with incidence angles of 20°, 30°, 40°, 50°, and 60° are shown with different colors. Compared with the measured multi-angle spectrum of olivine with corresponding incidence angles of 20°, 30°, 40°, 50°, and 60° (refer to Figure 4b), the simulated multi-angle spectral curves have well-established high consistency with the measured multi-angle spectrum.

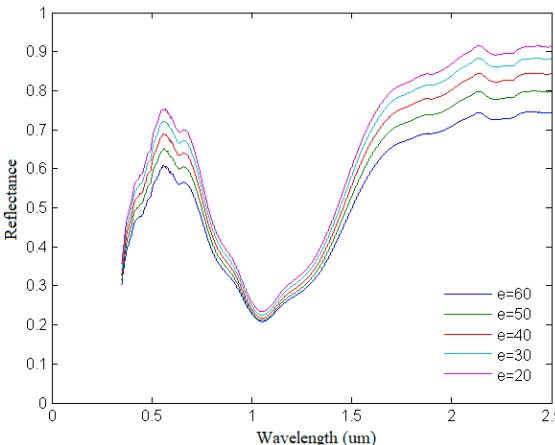

**Figure 10.** Simulated spectra of olivine.

4.1.3. Comparative Analysis between the Simulated Spectra and Measured Spectra of Olivine

To show the quality and accuracy of the photometric parameters retrieved from the multi-angle spectrum, a quantitative analysis between the simulated multi-angle spectrum and the measured multi-angle spectrum of olivine is performed (refer to Figure 11). Figure 11 is a comparison diagram of the simulated spectra (Figure 7) and measured spectra of olivine under different exit angle conditions with an incident angle of 0° and an azimuth angle of 210° (refer to Figure 4b). The root mean square error (RMSE) of each incidence angle between the simulated spectrum and the measured spectrum is calculated. The RMSE values for 20°, 30°, 40°, 50°, and 60° are 0.0067, 0.0328, 0.0380, 0.0363, and 0.0094, respectively. The results of the quantitative analysis show that the simulation effect is good, and the RMSE is of the order of $10^{-3}$–$10^{-2}$. The simulation results highlight the absorption characteristics of the main wavebands, indicating that the olivine photometric parameters obtained by optimized retrieval are more accurate and that the errors in the spectral simulation are small.

To further quantitatively analyze the dynamics of reflectance with different incidence angles between the simulated spectrum and the measured spectrum, the characteristic wavelengths located at 0.57 μm and 2.15 μm are selected. The comparison results are shown in Figure 12, where the reflectance of olivine at characteristic wavelengths of 0.57 μm and 2.15 μm with an incidence angle of 0° and azimuth of 210° is compared with that of the simulated spectra. The simulated spectrum well reflects the change in the reflectance as a function of angle, indicating that the photometric parameters of olivine obtained by optimized retrieval are more authentic.

*4.2. Retrieval of Photometric Parameters and Spectral Curve Simulation of Plagioclase*

4.2.1. Photometric Parameters Retrieval of Plagioclase

The photometric parameters of plagioclase, such as the SSA w and phase function parameters b and c, were retrieved from the mean spectral dataset, which were obtained by averaging the five multi-angle spectra with an incidence angle of 0° and azimuth of 180° and five multi-angle spectra with an incidence angle of 0° and azimuth of 240°. During the retrieval process, the phase function parameters are set with an upper and lower range—the c represents the amplitude of the phase function, and the b represents the front and back scattering ratio coefficients of the phase function. The SSA curve is continuous and smooth with distinct absorption features. The average values of the phase function parameters b and c are −0.6845 and 1.0000, respectively (Figure 13). In Figure 13b, negative values have more forward scattering, and positive values have more back scattering. When SSA, b, and c are simultaneously derived, the parameters SSA and b have a better performance than that of parameter c due to their great variation and less robustness. The most values of b inversions are negative, which reflects the forward scattering of silicate minerals. The

coefficient c reflects the amplitude of the phase function and characterizes the degree of side scattering, and the upper limit of 1 may be interpreted as stronger scattering. The RMSEs of the incidence angles of 20°, 30°, 40°, 50°, and 60° are 0.0178, 0.0118, 0.0178, 0.0122, and 0.0265, respectively.

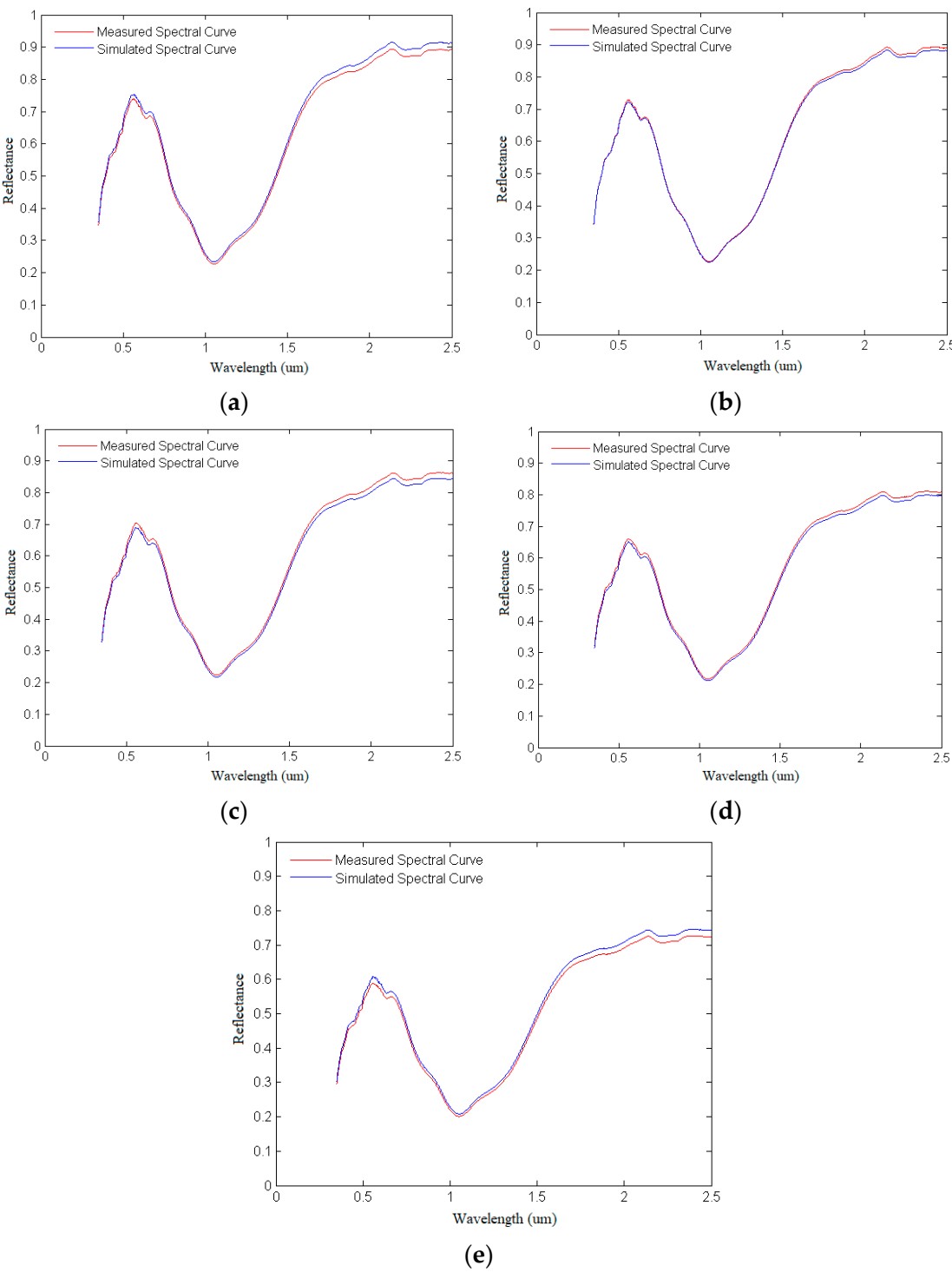

**Figure 11.** Comparison of the simulated and measured spectra of olivine at 20° (**a**), 30° (**b**), 40° (**c**), 50° (**d**), and 60° (**e**).

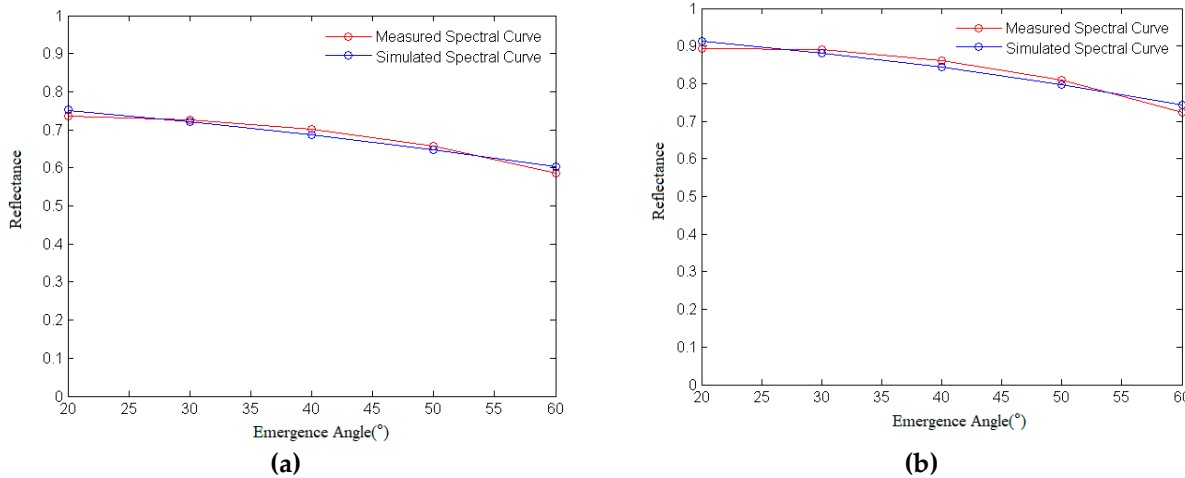

**Figure 12.** Comparison of the simulated and measured spectra of olivine at 0.57 μm (**a**) and 2.15 μm (**b**) as a function of angle.

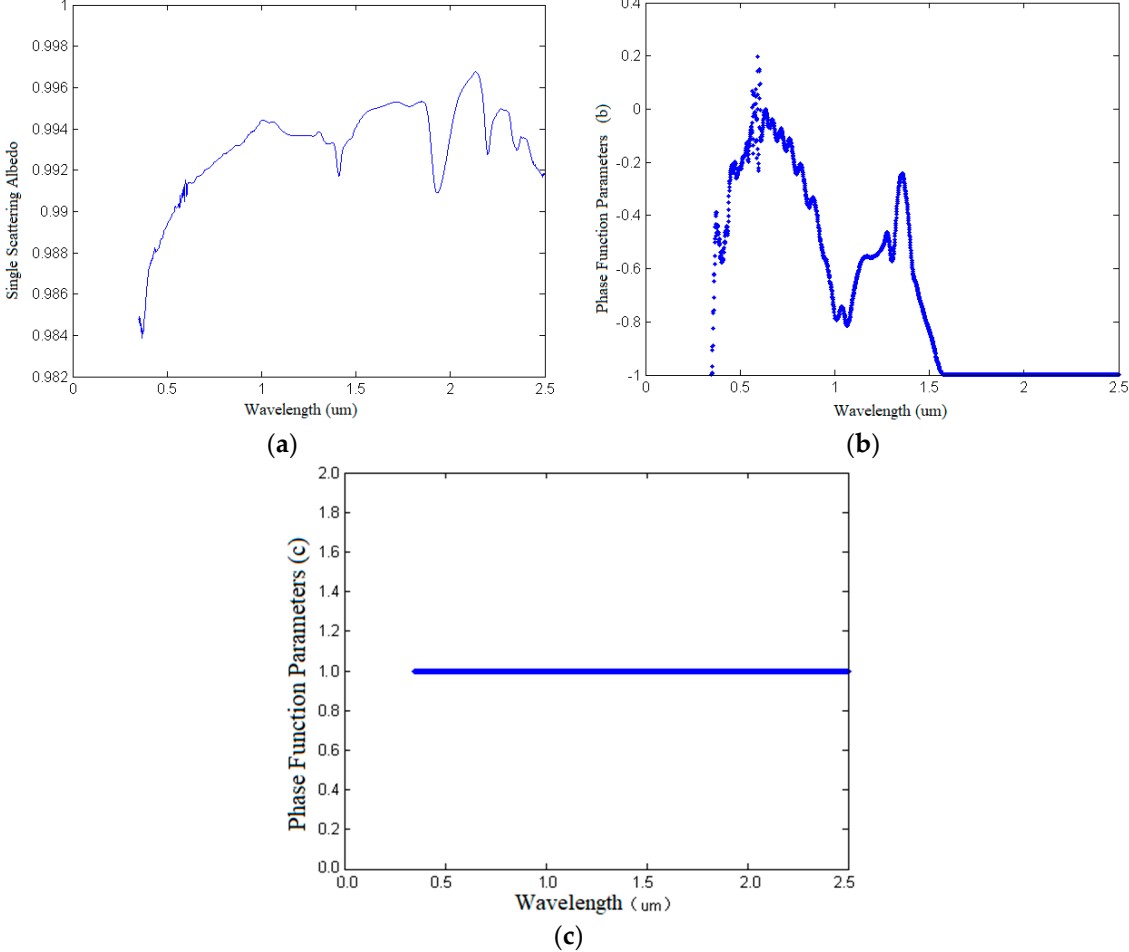

**Figure 13.** Photometric parameters of plagioclase: (**a**) SSA (w), (**b**) Legendre polynomial coefficient of the phase function b, and (**c**) Legendre polynomial coefficient of the phase function c of plagioclase.

### 4.2.2. Spectral Curve Simulation of Plagioclase Based on Retrieval of Photometric Parameters

Five multi-angle spectra with an incidence angle of 0° and azimuth of 210° were simulated by using the retrieved photometric parameters of plagioclase (refer to Figure 14).

In Figure 14, the multi-angle spectral curves with incidence angles of 20°, 30°, 40°, 50°, and 60° are shown with different colors. Compared with the measured multi-angle spectrum of olivine with corresponding incidence angles of 20°, 30°, 40°, 50°, and 60° (refer to Figure 5b), the simulated multi-angle spectral curves have high consistency with the measured spectrum.

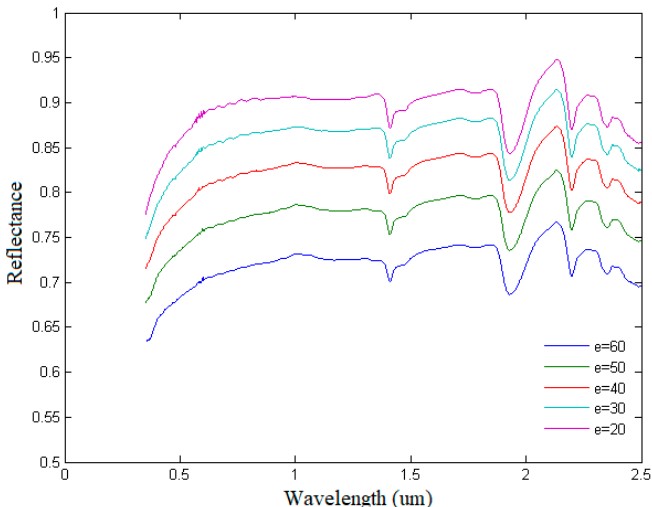

**Figure 14.** Simulated spectra of plagioclase.

### 4.2.3. Comparative Analysis between the Simulated Spectra and Measured Spectra of Plagioclase

To show the quality and accuracy of the photometric parameters retrieved from multi-angle spectra, a quantitative analysis between the simulated multi-angle spectrum and the measured multi-angle spectrum of plagioclase was performed (refer to Figure 15). Figure 15 is a comparison diagram of the simulated spectra (Figure 14) and measured spectra of plagioclase under different exit angle conditions with an incident angle of 0° and an azimuth angle of 210° (refer to Figure 4b). The RMSE of each incidence angle between the simulated spectrum and the measured spectrum is calculated. The RMSE values for 20°, 30°, 40°, 50°, and 60° are 0.0012, 0.0377, 0.0436, 0.0380, and 0.0035, respectively. The simulation worked well with an RMSE of $10^{-3}$–$10^{-2}$ orders of magnitude, and the absorption features of the main wavelengths were well highlighted. These results show that the photometric parameters of olivine obtained by optimized estimation have a high accuracy and can be applied in spectral simulations with small errors.

In Figure 16, the reflectance of plagioclase at characteristic wavelengths of 1.93 μm and 2.20 μm with an incidence angle of 0° and azimuth of 210° is compared with that of the simulated spectra. The simulated spectra can reflect the law of reflectance as a function of angle, which indicates that the photometric parameters of plagioclase obtained by optimized retrieval have high authenticity.

### 4.3. Retrieval of Photometric Parameters and Spectral Curve Simulation of Ilmenite of RELAB Spectral Library and Reliability Evaluation of Coefficient Setting of Legendre Polynomial of Scattering Phase Function

#### 4.3.1. Retrieval of Photometric Parameters of Ilmenite of the RELAB Spectral Library

In this paper, five multi-angle spectra of ilmenite (sample ID: MR-MSR-005) in the RELAB Spectral Library were used to retrieve the photometric parameters of ilmenite, namely, the SSA w and phase function parameters b and c. The noise of the SSA w is larger, which is mainly related to the lower spectral resolution (10 nm), but its absorption features are distinct. The average values of the phase function parameters b and c are −0.6775 and 0.5475, respectively (Figure 17). In our study, the results of the retrieval parameters of *b*, *c*, and *W* also showed a high correlation between b and c, but a relative low correlation between *b* and *W* comparing with the corresponding correlation of Sato et al. (2014) [35].

Sato et al. (2014) spatially resolved near-global Hapke parameter maps derived from LROC WAC, and they found the photometric properties of the lunar surface vary with wavelength and composition. They employed the Henyey–Greenstein double-lobed single particle phase function, but in our paper, the second-order Legendre polynomial equation is used. Both of them reflect the photometric parameters' variation with wavelength. In their study, the parameters w and b were calculated by least-squares fitting with 30 different starting parameter sets, and parameter c was derived by b using the empirical function based on the hockey stick relation (Hapke, 2012) [24]. In fact, we found that the results of Sato et al. (2014) showed that both of the parameter maps of b and c are highly correlated with w.

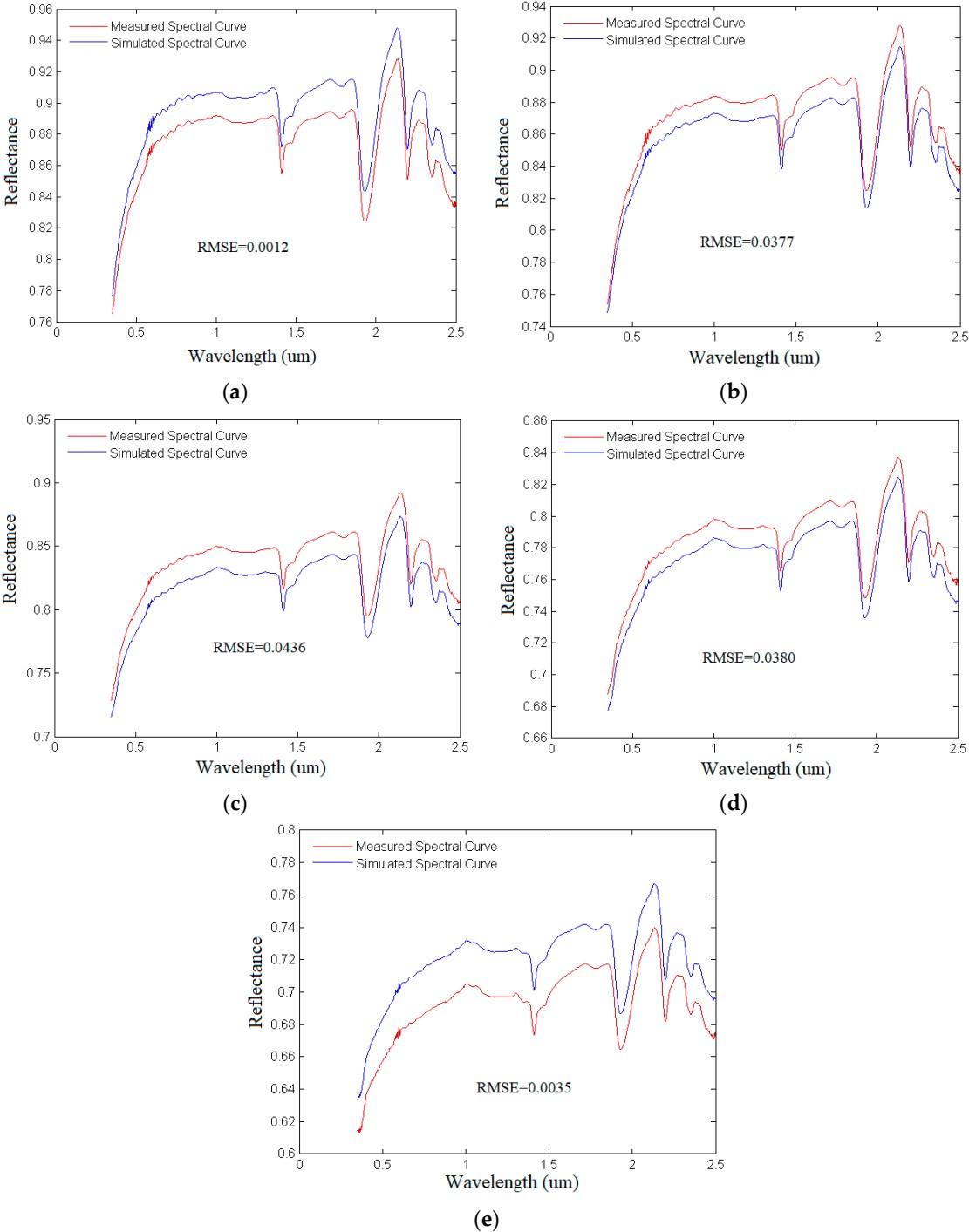

**Figure 15.** Comparison of the simulated and measured spectra of plagioclase at 20° (**a**), 30° (**b**), 40° (**c**), 50° (**d**), and 60° (**e**).

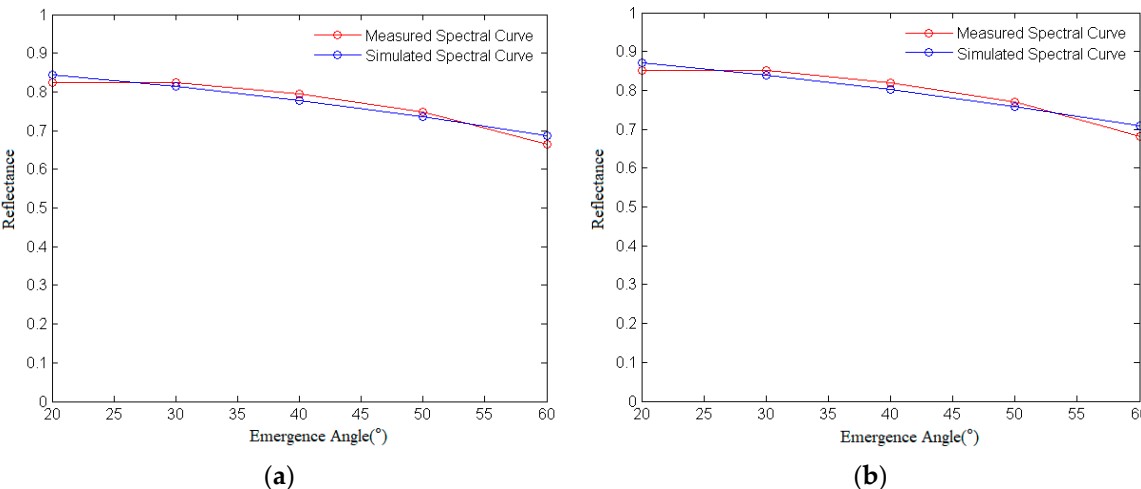

**Figure 16.** (**a**,**b**) Comparison of the simulated and measured spectra of plagioclase at some characteristic wavelengths as a function of angle.

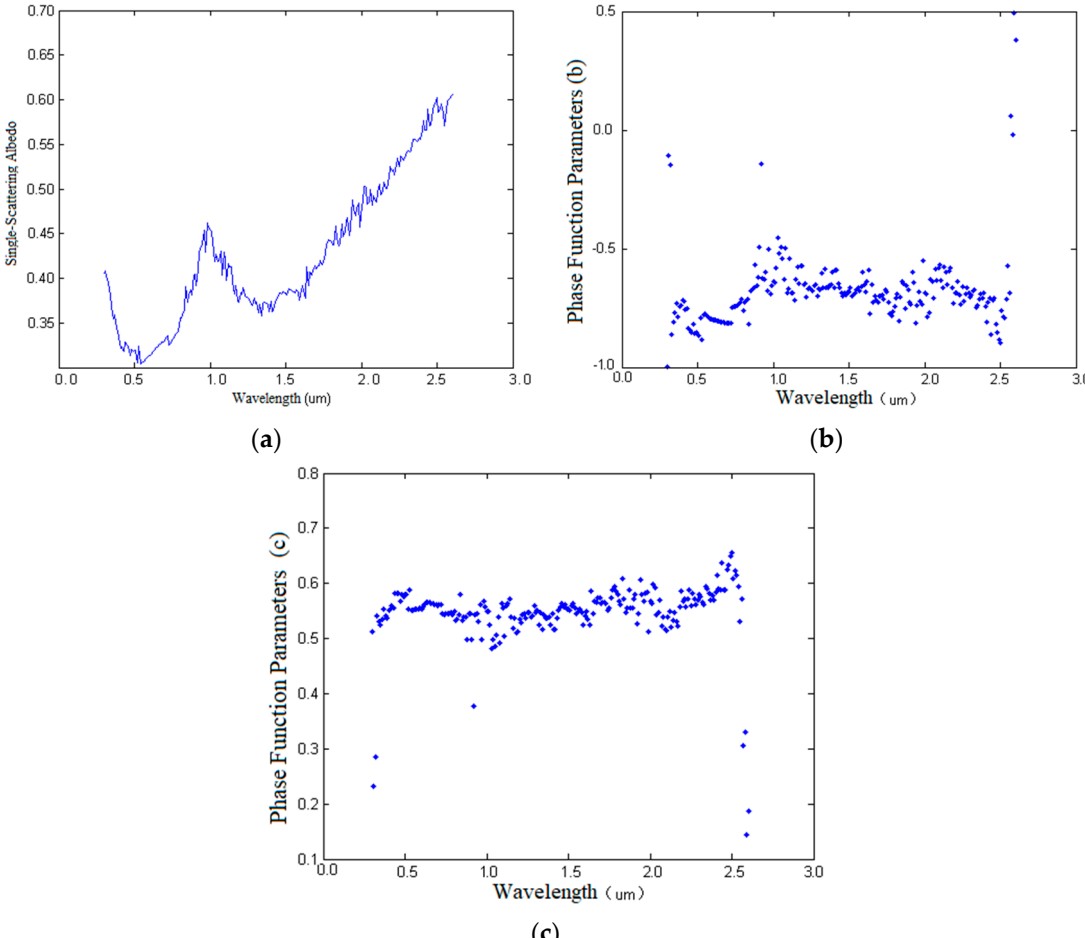

**Figure 17.** Photometric parameters of ilmenite: (**a**) SSA (w), (**b**) Legendre polynomial coefficients of the phase function b, and (**c**) Legendre polynomial coefficients of the phase function c of ilmenite.

### 4.3.2. Spectral Curve Simulation of Ilmenite of the RELAB Spectral Library Based on Empirical Values of Photometric Parameters b and c

In practical applications, phase function parameters b and c are often set by empirical values, and the SSA w is derived by using single-angle spectral data. Currently, the

Legendre polynomial coefficient of the scattering phase function in spectral simulations mainly refers to the approximate average value of the forward scattering coefficients of minerals proposed by Mustard and Pieters [22]. Generally, b is set to −0.4, c is set to 0.25 [9,36], and then the SSA w is retrieved from the single-angle spectra.

As the Legendre polynomial coefficients of the scattering phase function are difficult to measure, in this paper, the coincidence degree between the simulated multi-angle spectra and the measured values was analyzed as a function of the different parameters b and c and was utilized as an evaluation index of the reliability of the setting parameters.

The simulated multi-angle spectra with incident angles of 20°, 30°, 40° and 50° were compared with the measured multi-angle spectra (Figure 18). The contrast between the five simulated multi-angle spectra is very small, which does not truly reflect the effect of angle change on the reflectance spectra. Lucey's empirical parameters of the phase function are not fully applicable to the multi-angle spectra simulation of minerals [9].

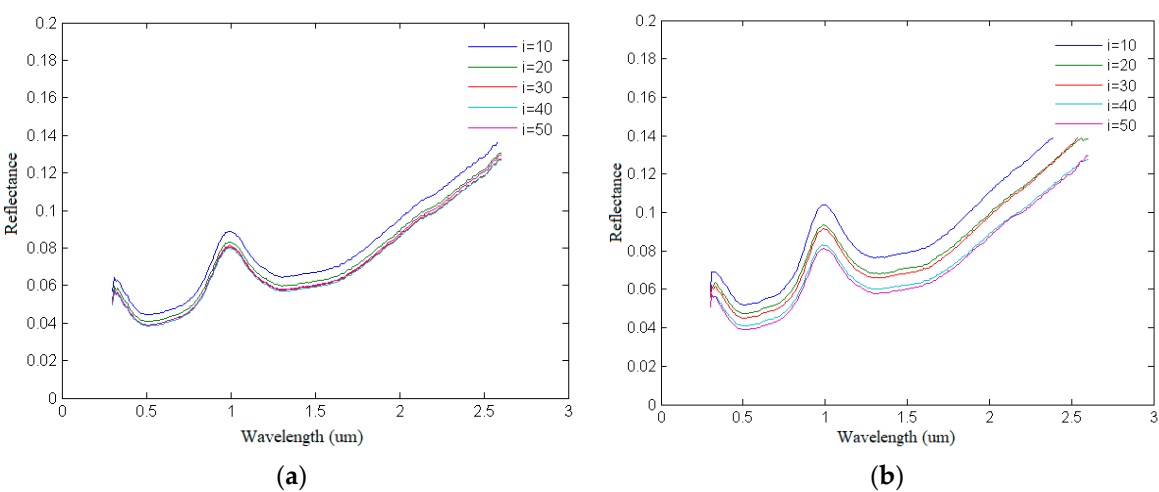

**Figure 18.** Comparison of the simulated (**a**) and measured spectra (**b**) of ilmenite when the Legendre polynomial coefficients of the phase function are set to −0.4 and 0.25.

### 4.3.3. Spectral Curve Simulation of Ilmenite of the RELAB Spectrum Library Based on the Average Value of Photometric Parameters of b and c Calculated from Multi-Angle Spectrum

The average value of the phase function parameters of each band retrieved from multi-angle spectral data (b = −0.6775 and c = 0.5475) was applied as the empirical value of the new phase function parameters, and the SSA w of ilmenite was retrieved again using single-angle spectral data. The five measured reflectance spectra were simulated, and the simulation accuracy was distinctly improved (Figure 19).

To further compare and analyze the influence of the empirical parameters of the phase function on the accuracy of the spectra simulation, variations in the reflectance of ilmenite at characteristic wavelengths of 0.52 μm and 1.00 μm with different incident angles were observed when both the emergence angle and azimuth angle were set to 0° (refer to Figure 20). As shown in Figure 20, the simulated spectral curves based on w, b, and c have a better correlation with the measured value among the three types of simulated spectral curves. This finding also showed that the photometric parameters derived from multi-angle spectral curves have a better performance and accuracy in simulating spectral curves.

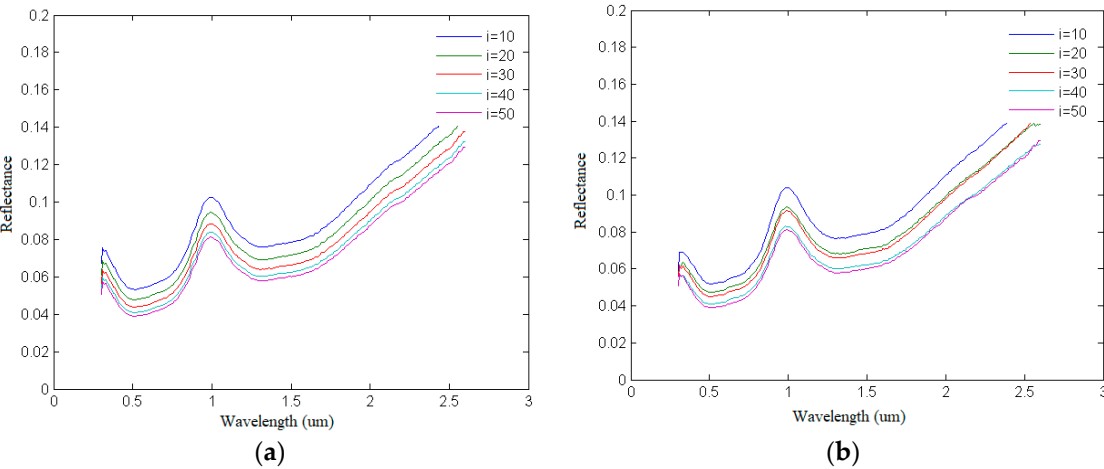

(**a**)　　　　　　　　　　　　　　　　　(**b**)

**Figure 19.** Comparison of the simulated spectra (**a**) and measured spectra (**b**) of ilmenite when the Legendre polynomial coefficients of the phase function are set to −0.6775 and 0.5475.

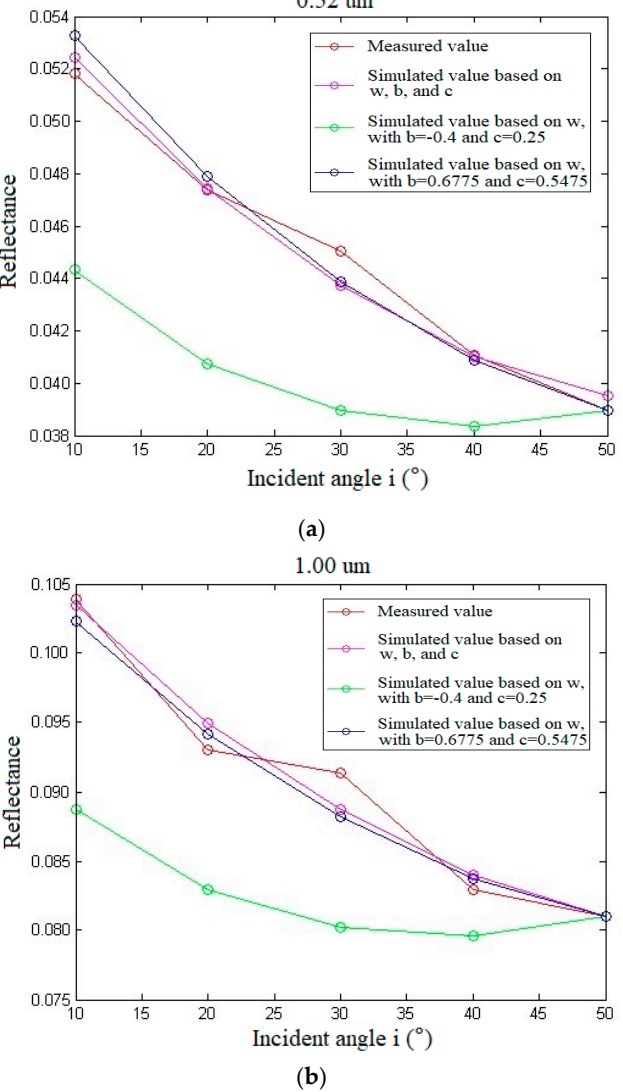

(**a**)

(**b**)

**Figure 20.** Comparison of the simulated and measured spectra of ilmenite at 0.52 μm (**a**) and 1.0 μm (**b**) versus angle.

In this paper, the photometric parameters w, b and c of ilmenite were calculated from multi-angle spectral data, and the measured spectra were simulated. The photometric parameter w of ilmenite was also retrieved from single-angle spectral data under the empirical value of the phase function, and the measured spectra were simulated. The errors of the spectra simulations using the two methods are compared in Table 2.

**Table 2.** RMSE comparison of the multi-parameter simulation, single-parameter simulation with the empirical parameters given by Lucey and single-parameter simulation with the improved empirical parameters.

| RMSE | e = 10 | e = 20 | e = 30 | e = 40 | e = 50 | Mean Value |
|---|---|---|---|---|---|---|
| multi-parameter simulation | 0.000436 | 0.001831 | 0.002696 | 0.001345 | 0.000345 | 0.001331 |
| single-parameter simulation (b = −0.4, c = 0.25) | 0.013568 | 0.009293 | 0.010363 | 0.00294 | $1.07 \times 10^{-9}$ | 0.007233 |
| single-parameter simulation (b = −0.6775, c = 0.5475) | 0.001701 | 0.001605 | 0.003035 | 0.001328 | $1.59 \times 10^{-8}$ | 0.001534 |

The error of estimation of the photometric parameters w, b, and c from multi-angle spectral data and simulation of measured spectra is the smallest, which is distinctly better than that from single-angle data under Lucey's empirical value of the phase function (b = −0.4 and c = 0.25). The former method truly reflects the effect of angle change on the reflectance spectra.

When deriving the photometric parameter w and simulating the measured spectra, it is better to take the average value of the phase function parameters of each band retrieved from multi-angle spectral data (b = −0.6775 and c = 0.5475) as the empirical value of new phase function parameters, which is superior to Lucey's traditional empirical value of the phase function (b = −0.4 and c = 0.25).

## 5. Conclusions

Optimized retrieval of the photometric parameters of the Hapke model can be realized by using the multi-angle spectral data. The derived results of the single-scattering reflectance w, which is the most important parameter in the spectra simulation, are stable. The estimation values of different wavelengths can be connected to continuous curves, which shows that the retrieval results have distinct physical significance and are not just a numerical fitting in a mathematical sense.

The estimated values of the Legendre polynomial coefficients of the scattering phase function at different wavelengths cannot be connected to very continuous curves, and there are mutations and steps at some wavelengths. As the measurement of the scattering characteristics of mineral particles is difficult to achieve, it is impossible to accurately judge whether these results reflect the real objective situation or are caused by the optimization retrieval error.

The setting of the Legendre polynomial coefficient of the scattering phase function mainly affects the simulation accuracy of the mineral spectra as a function of angle. From the existing measured data, the Legendre polynomial coefficient of the phase function based on the estimation of multi-angle spectra can simulate the law of mineral spectra as a function of angle.

The retrieved photometric parameters of minerals from multi-angle spectral data are stable and truly reflect the effect of the angle change on the spectra in the spectral simulations. However, due to the limitations in the testing conditions and workload, the RELAB, US Geological Survey (USGS) and other spectral libraries contain limited multi-angle reflectance spectral data, and only a few minerals have been tested at multiple angles. Therefore, to widely apply multi-angle reflectance, it is necessary to use samples of common minerals and carry out multi-angle spectral tests at a relatively high cost.

**Author Contributions:** All authors contributed equally to this work, and all the authors agreed to be co-first authors. All authors have read and agreed to the published version of the manuscript.

**Funding:** This research was funded by National Natural Science of Foundation, grant number 41701425, and by Major Projects of High-Resolution Earth Observation System (04-Y30B01-9001-18/20).

**Conflicts of Interest:** The authors declare no conflict of interest.

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
