# Peer review of "Retrieval of Photometric Parameters of Minerals Using a Self-Made Multi-Angle Spectrometer Based on the Hapke Radiative Transfer Model"

_remotesensing, doi:10.3390/rs13153022_

Round 1

Reviewer 1 Report

The authors have addressed all my concerns. I have no more comments and suggestions. I recommend this manuscript for publication in RS.

Author Response

The specific response to the reviewer's comments and suggestions can be seen in the attached file.

Reviewer 2 Report

Review of RemoteSensing 1255448 by Zhou et al.

June 21, 2021

This is a very useful study that mainly examines the effects of phase angle on deriving photometric parameters. This knowledge is important for accurate remote sensing of planetary surfaces, where phase angle is known to affect reflectance spectra and for which accurate correction is required for geological interpretation.

The need for this study is well done as is the explanation of optical constants and Monte Carlo modeling. These are often lacking in papers of this sort.

Minor comments:

Line 94: change “silicon Salt” to “silicate”

Line 104: delete “While”

Lines 115, 142: “Analytica” should be “Analytical” and ASD is now owned by Malvern.

Line 135: do not capitalize “Medium”

Line 146: should be Spectralon®; change “spectra Lab” to “Spectral Laboratory” data base”.

Figure 5: some of the boxes have spelling mistakes.

Line 306: do not capitalize “Under”.

Major comments:

Lines 112-122: what kind of incident light source did you use? Was its power stabilized? How often did you measure the Spectralon. Did you correct the spectra for dark current? Did you correct the spectra for irregularities in the 2-24 micron region of Spectralon? It doesn’t look like you did, which is OK, but should stated in the paper. How deep is the sample cup (to make sure you aren’t seeing through the sample to the bottom of the cup. How wide is the sample cup and the area seen by the spectrometer probe? (to make sure you don’t have spectral contributions from the side of the sample cup – especially at high angles). The viewed spot should be at least a couple of millimetres smaller than the size of the sample cup. How were the powders prepared? Did they have a flat and matte surface so that there are no “hot spots” in the spectra? Please provide these details in the method section. Some of this information is in lines 345-357, so should be moved up to lines 112-122. On line 255-356: when you say 10 sets of data, do you mean that you acquired 10 spectra of the target that you then averaged to come up with a single spectrum?

Section 3. Did you check to see whether moving the probe causes spectral artifacts to appear. In our laboratory we have found that when you move or wiggle the fiber (even without changing the phase angle) the spectra can vary because changing the position of the fiber causes changes in internal reflections and polarizations.

Lines 150-151: while average grain size is important, the range of grain size is even more important for deriving optical constants from reflectance measurements. Did you measure the maximum and minimum grain sizes? What were they?

Lines 328: what program did you use for the Newton interpolations?

Sample spectra: it is important to have pure unaltered minerals for deriving optical constants and spectra that are properly processed. To do this you need to correct the spectrum of the Spectralon you used to that of a 100% reflector (LabSphere or ASD provide such correction files). I also noticed that the plagioclase spectra have many absorption features that indicate that the sample is either not pure plagioclase or weathered (e.g., around 1.4, 1.9, 2.2 and 2.3 microns). You should mention that optical constants in these regions would be incorrect for pure plagioclase.

Line 283: in your solving for the optical constants what initial value did you use for the real part of the optical constant for each mineral? How did you come up with those initial values? Did you allow n to vary during interpolation?

Author Response

The specific response to the reviewer's comments and suggestions can be seen from the attached file.

Reviewer 3 Report

The manuscript used multi-phase observations and the Hapke model to derive the photometric parameters of certain mineral samples, and validated these parameters by comparing with real observations of other samples.

Merits:

  • Potential demonstration of multi-phase observations and photometric parameters retrieval based on a relatively simple setup.
  • Potential extra test of the Hapke model fitness of specific minerals.

Demerits:

  • Multi-phase observations for retrieving photometric parameters are common (e.g., see photometry for the Moon and asteroid, etc.).
  • The observation data have 0 deg incidence and 20-60 deg emission, translating to 20-60 deg phase angle. Therefore, they did not cover potential forward-scattering of the samples (90-180 phase angle). Such observations can be done by increasing incidence angle (i.e., tiling down the light source).
  • Using only Lucey’s values for comparison can be inappropriate.
  • Some claims were not supported with references.
  • The abstract was incomplete and cannot convey the important messages and findings.
  • The manuscript was severely fragmented and flawed, and the logic flow is confusing, I cannot closely follow what the authors were trying to say. The current form of the manuscript, and the data it presented, cannot facilitate any meaningful discussion.
  • Inconsistent use of words. Some variables in equations were not explained.
  • The manuscript gave a strong impression that the authors did not pay reasonable efforts in proofreading and editing the manuscript. The authors simply copy-and-paste the comments of other reviewers to their manuscript as-is, without any considerations about the context.

Suggestion to authors:

  • Try to standout your multi-phase method in a global context, and/or to extend its application range so that your contribution can lead to some interesting findings.
  • More complete experimental datasets are needed, especially the forward-scattering observations.
  • Compare to not only Lucey’s values, but also others. It is impossible to only consider Lucey’s values as globally valid for all minerals. If that is the convention, then the authors need more references to support such claims.
  • Some more understanding about the Hapke model, the phase function, and their naming conventions would be good. Hapke (2002, Icarus) can be a good short-cut.
  • Carefully edit and proofread the manuscript, consider sending it to the editing company again after any substantial revisions.

Author Response

(The authors gave the same response as above.)

Reviewer 4 Report

In this paper, the authors proposed a self-made measurement system that allows for capturing the multispectral characteristics of the scanned materials. Although the authors did an interesting work, my main concern is that it does not necessarily fit the scope of the Remote Sensing journal, and I think that it might be targeted to a different audience. To this end, I encourage the authors to resubmit this work to another, perhaps better-fitted journal, e.g., Sensors (https://www.mdpi.com/journal/sensors).

Author Response

(The authors gave the same response as above.)

Round 2

Reviewer 3 Report

Thank you for the authors' efforts in improving the manuscript and responding to my questions. I will give a few quick notes below.

The authors mentioned several important points in justifying their results and methodology, including how they modified the phase function, and why increasing incidence angle is not feasible for their setup. They should be included in the manuscript explicitly.

 Judging from the photos of the equipment, I am still not convinced that the authors cannot tilt down the light source slightly (10~45 deg perhaps). Therefore, one way for the authors to defend themselves would be to include a small section presenting, with figures, how bad the observations can be under a tilted light source. However, I still encourage the authors to obtain new measurements under various angles.

The authors mentioned in (p.3, ln 127) that "the incident angle between the probe and the light source varies between 40and 90o;...". Incidence angle usually refers to the angle between the light source and the normal vector of the surface, not the probe. Moreover, does that mean the authors' setup can indeed increase the incidence angle? Please explain and clarify in the manuscript.

I believe the results presented in the manuscript may draw questions from the readers, especially the estimation of parameters b and c (fig. 8b, c and fig. 12b, c). This is because these curves are too unnatural with apparent under- and over-cutting (fig. 12c is even a constant) and they showed signs of undesirable estimation. Therefore, the authors should carefully address and explain such issues in the manuscript.

Some more discussions besides Lucey's constants will be better. For example, the authors may discuss their results with the overall lunar photometric properties presented by Sato et al. (2014).

Consider moving section 3 into a subsection in section 4. This is because data collection is part of your overall workflow. By doing so, the readers can read the manuscript's flowchart first to gain an overview of what the authors were doing.

Author Response

The response to the reviewer can be found in the attached file.

Thanks a lot for spending so much effort and time on our manuscript, we're always appreciated your valuable comments and suggestions.

Reviewer 4 Report

Although I had concerns on the suitability of this manuscript to Remote Sensing, I ultimately agree that the ideas might be indeed interesting for the RS readership. Overall, the paper is well-written and sound, and it could be considered for publication.

Author Response

(The authors gave the same response as above.)
